## PROCEEDINGS A

atmospheric science, geophysics, meteorology

vertical winds, thermosphere, particle transport

**Author for correspondence:**
D. J. Brener
e-mail: daniel.brener@ed.ac.uk

# On the force of vertical winds in the upper atmosphere: consequences for small biological particles

A. Berera and D. J. Brener

The Higgs Centre for Theoretical Physics, James Clerk Maxwell Building, Peter Guthrie Tait Road, Edinburgh EH9 3FD, UK

AB, 0000-0001-5005-7812; DJB, 0000-0003-3340-0177

For many decades, vertical winds have been observed at high altitudes of the Earth's atmosphere, in the mesosphere and thermosphere layers. These observations have been used with a simple one-dimensional model to make estimates of possible altitude climbs by biologically sized particles deeper into the thermosphere, in the rare occurrence where such a particle has been propelled to these altitudes. A particle transport mechanism is suggested from the literature on auroral arcs, indicating that an altitude of 120 km could be reached by a nanometre-sized particle, which is higher than the measured 77 km limit on the biosphere. Vertical wind observations in the upper mesophere and lower thermosphere are challenging to make and so we suggest that particles could reach altitudes greater than 120 km, depending on the magnitude of the vertical wind. Applications of the larger vertical winds in the upper atmosphere to astrobiology and climate science are explored.

## 1. Introduction

Vertical winds up to $100\,\mathrm{m\,s^{-1}}$ have been observed in the upper mesosphere and thermosphere layers of the Earth's atmosphere for many decades. These vertical winds are observed to be sustained for of the order of minutes to an hour. High-altitude measurements of vertical winds have limited temporal, spatial and vertical coverage. Ground-based measurement techniques using optical imaging are restricted to narrow altitude ranges but have a greater temporal coverage. *In situ*

**Table 1.** Summary of the variation in extreme vertical wind observations with altitudes of interest, and example supporting studies. Some durations are rough estimates from the figures in the original papers.

| $z$ (km$^{-1}$) | range of maximum $w$ (m s$^{-1}$) | duration (min) | reference |
|---|---|---|---|
| 240 (F-region) | 138–150 | 15–120 | [3–5,7] |
| 120–130 (E-region) | 30–42 | 15–25 | [2,4,17] |
| <110 | 10–32 | 20 | [18] |
| 103 | 10–50 | 30 | [15] |
| 86 | 65 | 20 | [19] |

measurements, using rockets to disperse trackable tracer gases, are temporally limited, but are able to sample larger altitude ranges. Reviews of vertical winds in the thermosphere are given by Smith in 1998 [1] and Larsen & Meriwether in 2012 [2].

Owing to the Earth's geomagnetic field, particles in the solar wind are channelled such that they penetrate the atmosphere in the polar regions. The charged particles collide with the neutral air, transferring their energy and momentum. The transfer of momentum and subsequent heating at these altitudes is understood to cause some of the largest observed vertical winds. The mean vertical wind in the polar regions at these altitudes is typically of the order of a few tens of m s$^{-1}$. However, upward vertical winds of up to 150 m s$^{-1}$ have been observed [3–5] at around 240 km. At polar regions where the aurora occur, known as the auroral zone, vertical winds have been observed with magnitudes around 50 m s$^{-1}$ [6–15]. Some studies have reported vertical winds of more than 100 m s$^{-1}$ [3,4,16]. Most of these observations are of vertical winds at altitudes of around 250 km, but some are as low as 90 km (e.g. [15]) and they are usually seen during geomagnetic storms. It must be emphasized that vertical winds in the mesosphere of more than 100 m s$^{-1}$ have not been observed so far. In table 1, we have included a summary of key studies reporting large vertical winds at different altitudes.

In the last year, a new set of observations have reported a variance in vertical velocities as high as 60 m s$^{-1}$ in the mesosphere [19]. Although these values are smaller than those in the thermosphere, they are more than $5\sigma$ larger than the variability in normal vertical velocities. These extreme vertical winds were observed when the ionosphere was quiet for a few hours and at an altitude too low for plasma instabilities to be generated. This led the authors to suggest that the origins of the winds were not related to auroral activity. There are also new observations that show unexpected signatures of metallic ion layers at altitudes as high as 180–200 km [20]. These observations indicate that non-auroral activity might be involved in the transport of particles deeper into the thermosphere in a combination of both neutral vertical winds and geomagnetic Lorentz forcing [20].

Little theoretical progress has been made in understanding the large vertical wind observations [2,21; K Garcia-Sage 2020, personal communication]. Deviations from the hydrostatic balance have been studied (e.g. [22–24]), but they do not last long enough to create vertical winds sustained over several hours [2]. Fully non-hydrostatical models (e.g. the global ionosphere–thermosphere model [25]) are unable to reproduce the large vertical winds because they typically run at too low a spatial and temporal resolution [K Garcia-Sage 2020, personal communication; J Yue 2020, personal communication]. This is not to say that the vertical winds in the current operational models are wholly unrealistic, but rather that there exists an observational literature of larger vertical winds which have yet to be fully captured by the models (see review paper [2], which came after [22]). The physical mechanism generating the vertical winds is outside the scope of this paper—all that matters is the observations of such strong vertical winds.

Various measurements have shown that there are particles of a radius around the size of a micrometre and in reported concentrations of approximately 1 particle cm$^{-3}$ in the stratosphere. [26–30]. Additional studies have found that these particles include bacteria [28,31,32]. The highest altitude that biological particles have been found is 77 km [33]. These were fungal spores, with a

radius within an order of magnitude of a micrometre. Bacteria have been found as high as 41 km [28,32]. However, these are likely to be underestimates, as the studies were very limited because of the technical difficulty of growing cultures of the bacteria and fungi captured in sealed capsules from rockets sent to these altitudes.

More recently, cosmic dust samples from the surface of the International Space Station (ISS) were found to have DNA from several kinds of bacteria which were genetically similar to those found in the Barents and Kara Seas' coastal zones [34]. The investigators hypothesized that the wild land and marine bacteria DNA could transfer from the lower atmosphere into the ionosphere–thermosphere using the ascending branch of the global electric circuit or the bacteria found may have had a space origin. This paper highlights that there is a body of evidence of large vertical winds in the upper atmosphere which have the capability to push larger particles higher, hence potentially extending the biosphere further than previously thought. Such events are likely to be fluctuations and therefore rare occurrences, but, over millennia, they may have astrobiological significance.

We are aware that the idea that such biological particles may exist in the upper mesosphere, where the density is usually insufficient to support them for long, is slightly contested in private discussions. However, no studies have been conducted that provide alternative explanations for the observations. For our work, it is sufficient to say that such occurrences are probably very rare, but on geological time scales may be significant.

A mechanism known as gravito-photophoresis, arising from irradiation of particles by sunlight, has been shown to elevate micrometre-scale particles to altitudes of approximately 83 km [35]. This mechanism has been little investigated with only a handful of papers examining its effects on the upper atmosphere (e.g. [35,36]).

The existence of noctilucent clouds at altitudes of 80–100 km provides evidence that small dust-like particles can be found there [37]. These clouds are most often observed closer to the poles or latitudes greater than 50°, and so are also known as polar mesospheric clouds. A definite source of the condensation nuclei is interplanetary space dust from meteors and passing comets. These are known to release large quantities of dust into the upper atmosphere [38–40].

However, it is also possible that the dust particles are terrestrial in origin. Volcanic eruptions are terrestrial events capable of significant upward thrust, projecting ash into the stratosphere [41,42]. Modelling of the 1883 Krakatoa eruption indicates that dust from the volcanic ash cloud diffused up to around 85 km [43,44]. This idea is supported by observations of noctilucent clouds that appeared at the time of the eruption [45]. A recent study found that particles of size 50–100 nm could be projected in volcanic eruptions to the upper mesosphere, and particles of size 10 nm to more than 120 km [46]. Furthermore, with increasing numbers of commercial spaceflights, especially now including space tourism, one would reasonably expect to see increases in the incidences of short-lived biological contamination of the mesosphere–lower thermosphere [47,48].

There is a growing interest in the upper atmosphere from the astrobiology community, but there has been no concerted research effort placed on the mesosphere and thermosphere [48,49]. The recent investigations and publicity around the atmosphere of Venus have also instilled interest in the possibility of life transfer from and to nearby planets [50–52].

In the last decade, work has been done to extend global numerical weather prediction models into the thermosphere (e.g. WACCM in the USA [53], CMAM in Canada [54] and the Met Office Unified Model in the UK [21]) and whole atmosphere modelling studies are considering the role of the mesosphere and thermosphere in the Earth's climate [55–57]. Although these models are non-hydrostatic, which allows for the vertical acceleration necessary for vertical winds, there are presently no successful modelling studies of the larger vertical winds reported in the observational review papers.

Horizontal winds are, however, well captured by these models. Hence, particle transport in the horizontal is well understood through modelling of Lagrangian coherent structures (e.g. [58]), but these studies assume that the vertical motion is *negligible*. Owing to the strengths of the vertical winds needed to suspend or propel a large heavy particle upwards, we were motivated to take

an unconventional, yet simpler approach to making some first estimates of the maximum altitude attainable by such a particle.

The purpose of this paper is to illustrate the approximate strength of these extreme winds, which has implications for transporting particles larger than molecules higher than experiments have yet measured. We recognize that the natural horizontal symmetry of upper atmospheric particle transport can be broken by the observed large fluctuations in vertical winds, allowing for the complicated horizontal dynamics to be ignored to a good approximation during such events. This provides a clear justification to use a one-dimensional model.

The technicalities of this assumption are left for later discussion in this paper, but, in summary, large vertical winds have been correlated with horizontal winds in auroral arcs which may enable a particle to remain in a strong vertical wind flow. This is just one mechanism, and there are likely to be other such processes working at smaller scales. We hope that this short proof-of-concept paper will act as a precursor to future studies using sophisticated three-dimensional (3D) global numerical weather prediction models to reproduce the vertical winds. By considering the possibility of larger particles existing at altitudes higher than previously studied, we open up interesting applications in other fields such as astrobiology and climate science, which we discuss in the penultimate section.

This paper presents a theoretical argument for the presence of heavier nanometre-sized or larger particles in the mesosphere and lower thermosphere. To date no dedicated field campaigns have been conducted to search for such particles at these altitudes. By combining observations of strong vertical winds at these altitudes with some theoretical arguments, we show that there is a strong case for the presence of such particles at these high altitudes at intervals dependent on the frequency of strong vertical wind events.

## 2. Theory

### (a) One-dimensional model

Empirically speaking, the simplest approximation for the upper atmosphere circulation pattern (i.e. zonal and meridional mean flow) is that these are roughly the same over large regions of the Earth, in magnitude and variation, driven by tides and waves. Because of solar particle interactions, the poles are more exceptional, but the approximation remains largely true to zeroth order. Hence, as has been shown in thermosphere tracer particle transport studies, horizontal winds will move a particle a considerable distance around the globe [58]. However, as has been noted from vertical wind observations around the world (see review [2]), the vertical forces that a particle experiences from vertical winds remain similar.

By modelling the particle transport in only the vertical direction, our model assumes that the profile of the vertical wind at different horizontal regions above the Earth is to a good approximation similar. This can be seen in the observational literature. Though taken at different locations over the Earth, the vertical wind patterns are approximately the same, with periods of the order of hours of upward and then downward motion and with velocities ranging up to tens of metres per second [2].

Thus a test particle may be blown around considerably in the horizontal direction; however, in whatever region it is in, the vertical component it experiences has a similar behaviour, but weaker strength. If one were just interested in the vertical motion of this test particle, its dynamics looks somewhat like a random walk, in that it initially may experience upward vertical wind, and then be blown horizontally to a region where it may experience either upward or downward wind and so forth.

If all one is interested in is the vertical motion of the test particle, independent of where it lies horizontally, then one could describe its vertical motion through simply a one-dimensional differential equation. The similarity of the vertical wind over different horizontal regions implies an approximate horizontal translational symmetry that can be factored out of the motion. This provides a significant simplification of the problem.

The resulting one-dimensional equation should then be stochastic, with the randomness describing the instances when the test particle feels upward versus downward wind as well as variations in wind speed. Modelling such an equation is currently too big a first step. As a first step, we would just like to assess the strength of vertical wind forces on a test particle and how high such a test particle might traverse. With this in mind, the first most basic question is how strong a vertical wind is needed to overcome the downward gravitational force. Here, we will develop a simple differential equation to express this modest goal. The equation itself is not new and has been applied in a different context, as it simply expresses, for a directed flow, the terminal velocity a test particle can achieve. However, this equation provides a starting basis for further development, whereby it can be extended into a stochastic equation with more detailed description of drag forces.

For example, our analysis in the next section will go beyond just this equation and include specific cases for auroral arcs over the polar regions. Such details might eventually be understood on aggregate and compactly expressed within a single stochastic equation, but, at this early stage, we have little quantitative understanding of how to model all such probabilistic events. This is the first paper to identify this process of nano- to micrometre test particles being propelled upwards by vertical wind, and we are simply exploring the physical attributes of this process.

### (i) Continuum approximation

We will consider test particles climbing from the upper mesosphere, at around 90 km, into the thermosphere. At these high altitudes, the density of the air is five orders of magnitude lower than at the surface. In most situations, 3D particle transport is characterized by the ratio of the fluid's mean free path to the radius of the test particle that is to be transported; this is known as the Knudsen number (Kn). The equilibrium mean free path of the air at 90 km is approximately 0.02 m, so for a micrometre-sized test particle Kn is 23 700 [59]. Being much greater than unity, this implies that the dynamics of such a test particle is governed by the non-continuum, kinetic regime. This can also be noted by calculating the Reynolds and Mach numbers for the flow, which are dominated by the very low densities at these altitudes.

However, in the case of a strong directed one-dimensional (1D) flow, such as what has been observed of the vertical winds at these altitudes, to zeroth order the forces in the continuum limit can be applied. We approximate the situation as an entirely vertical directed flow, analogous to an atmospheric river. This point of view is motivated by the fact that the observations show these long-lasting streams of upward flow. Effectively, we neglect the kinetic fluctuations, which are what dominate the free molecular flow/kinetic regime, characterized by $Kn \ll 1$.

If we were to include such fluctuations it would be necessary to consider the mechanics as a three-dimensional Brownian movement within the directed flow, which is unnecessary detail for this paper's objective. We note that there is a substantial body of literature which discusses the various limits of applicability of the continuum approach at high Kn number (e.g. [60]). It is also worth noting that random lateral fluctuations will be on the scale of the mean free path, and thus smaller than the transport distance of the bulk, i.e. the horizontal winds are much more significant.

Furthermore, in engineering, where this problem is encountered for microchannels that have a large aspect ratio (width-to-height), it is conventional to use spanwise space averaging to define an averaged velocity profile, hence defining the equivalent macroscopic quantities [61]. This is equivalent to our use of the vertical velocity principally because we neglect in these simple calculations the horizontal motions.

### (ii) Threshold velocity for a test particle to ascend with vertical wind

Our study will focus on a fiducial test particle that has a characteristic size and mass larger than the surrounding constituents, i.e. atoms and molecules, of the atmosphere at that altitude. In describing how a test particle is carried by the wind, two forces shall be considered. The first is the weight of the test particle due to its mass. The second is the force carried by the momentum of

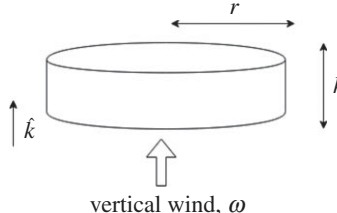

**Figure 1.** Test particle as a disc, orientated such that the vertical wind, a continuous directed flow, impacts on its major axis, pushing it upwards.

the vertical wind, distributed over the surface of the test particle. Using this picture, we obtain the force of a vertical wind acting on the particle. The form of this force is unsurprisingly the same as the standard drag force on a particle falling at terminal velocity, only now acting in the opposite direction to propel the particle.

We model the test particle as a disc with a drag coefficient of unity, for simplicity. This means that all the upward vertical wind hits the test particle at the same time. No significant part of the test particle feels more wind against itself, thereby reducing the problem to one dimension. It is also helpful as it mirrors the most abundant measurements of vertical winds in the upper atmosphere as instantaneous winds at discrete altitude points.

Let the test particle be a disc of radius $r$ and thickness $h$, as illustrated in figure 1, and let it sit with its major axis perpendicular to the vertical wind direction. So the density of such a test particle, $\rho_p$, can be written as

$$\rho_p = \frac{m}{\pi r^2 h}, \tag{2.1}$$

where $m$ is the mass, $r$ is the radius and $h$ is the thickness of the test particle. Consider a vertical wind blowing upwards with velocity $w$, which impacts on the circular disc surface. Then the mass of the air which will impact on the base surface of the test particle in some finite time $t$ is

$$M_{\text{air}} = \rho \pi r^2 w t, \tag{2.2}$$

where $\rho$ is the density of the air in a cylindrical volume element.

The force due to the vertical wind is essentially the same as that due to the continuum drag force, but acting in the opposite direction, so the test particle can be considered to be moving *with* the flow of the wind after some finite time $t$, to allow the wind to have sufficient force on the test particle. Thus air resistance, against the direction of the test particle's motion, shall be henceforth neglected.

Let the velocity of the vertical wind, $\vec{w}$, be upwards, perpendicular to the Earth's surface ($\hat{k}$) so that $\vec{w} = w\hat{k}$. Assume that the vertical wind is constant with both altitude and time. This is a good approximation over some period of time usually lasting tens of minutes and over some distance extending over several kilometres, when there is a constant vertical wind. Let the velocity of the particle moving upwards with the wind be $\vec{v} = v\hat{k}$. Hence the velocity of the wind relative to the particle is $(w - v)\hat{k}$. Since all the motion is restricted to one dimension, the $\hat{k}$ unit vectors shall be dropped from now on, with upwards defined as positive.

From equation (2.2), the mass of the air impacting on the lower surface of the test particle in some time $t$ is

$$M_{\text{air}} = \rho \pi r^2 (w - v) t, \tag{2.3}$$

which gives the momentum transferred by the wind to the test particle,

$$p = \rho \pi r^2 (w - v)^2 t. \tag{2.4}$$

Hence the force on the test particle due to the wind is given by

$$F_{\text{wind-particle}} = \frac{\mathrm{d}p}{\mathrm{d}t} \tag{2.5}$$

$$= \rho \pi r^2 (w - v)^2. \tag{2.6}$$

The equation for the two forces, the weight and vertical wind momentum, is therefore

$$m \frac{\mathrm{d}v}{\mathrm{d}t} = -mg + \rho \pi r^2 (w - v)^2. \tag{2.7}$$

As mentioned earlier, this equation is equivalent to the equation for the terminal velocity of a test particle with an opposing drag force proportional to the quadratic vertical wind speed, where the drag coefficient $C_d$ in the quadratic drag law is unity. This occurs when the entire wind flow onto the bottom of the disc comes to rest, creating stagnation pressure.

As the relative velocity force term in the above equation is squared, the force due to the vertical wind cannot change sign. So in the regime where the vertical wind blows downwards, meaning $\vec{w} < \vec{v}$, the equation at the moment wrongly suggests that the test particle will still move upwards. Put simply, taking the limit where $\vec{w} \to -\infty$, the acceleration of the test particle will remain in the positive $\hat{k}$ upward direction. To give the force the correct sign, the Heaviside unit step function, $H$, is introduced as

$$H(w) = \begin{cases} 1, & \text{if } w > 0, \\ -1, & \text{if } w < 0, \end{cases}$$

Using equation (2.1), and introducing the Heaviside step as above, equation (2.7) can be written as

$$\frac{\mathrm{d}v}{\mathrm{d}t} = -g + H(w - v) \frac{\rho(z, t)}{\rho_p h} (w(z, t) - v(t))^2. \tag{2.8}$$

Physically, this equation says that if the force due to the vertical upward wind is strong enough, it can overcome the force due to the test particle's weight, and cause the test particle to accelerate upwards. This equation is separable, yielding

$$t = \frac{1}{g} \int_0^{v(t)} \frac{\mathrm{d}v'}{b(w - v')^2 - 1}, \tag{2.9}$$

with the boundary conditions that in some time $t$ the test particle reaches a velocity $v(t)$, having been at rest initially, and $b = H\rho(z)/gh\rho_p$. Solving this integral by substitution, and then rearranging, one finds that

$$v = w - \sqrt{\frac{g}{\lambda}} \left( \frac{1 + \left( \left( \sqrt{(\lambda/g)}w - 1 \right) / \left( \sqrt{(\lambda/g)}w + 1 \right) \right) \exp\left( -2t\sqrt{\lambda g} \right)}{1 - \left( \left( \sqrt{(\lambda/g)}w - 1 \right) / \left( \sqrt{(\lambda/g)}w + 1 \right) \right) \exp\left( -2t\sqrt{\lambda g} \right)} \right), \tag{2.10}$$

where $\lambda = H(w - v)(\rho(z, t)/\rho_p h)$. It is assumed that $t$ is sufficiently small such that the air density $\rho(z)$ and vertical wind speed $w$ remain constant. In the limit that $t \to \infty$, we obtain the correct steady-state solution, as found in equation (2.12). When $t = 0$, we find $v = 0$ as expected. From this equation, one finds that the time scale to reach terminal velocity is approximately $\sqrt{1/\lambda g}$. For the standard test particle that we consider (see §2b), this comes out at less than a second, which is shorter than the time scale of the observed large vertical winds, hence enough acceleration can be provided.

The Heaviside function is introduced to maintain some stability when working with real observations which can sometimes have cases where $w < 0$, but really this equation is only valid for the cases where $v < w$ and $w > 0$.

Let us imagine the situation where initially the test particle is at rest, and then some upward wind blows against it. Provided the wind is stronger than the test particle's weight in the air, the test particle will start to move upwards, accelerating in some finite time $t$ to reach the steady state. This is when the wind just balances the weight of the test particle, and at this point the

test particle's velocity will approach, but generally not reach, the speed of the wind. To see this, consider the limiting cases of the parameter $\lambda$ when the wind is blowing upwards such that $H(w - v)$ is positive in equation (2.10). When the parameter $\lambda$ is maximized the test particle's velocity is closer to that of the wind.

Owing to the disc geometry chosen, $\lambda$ is maximized when the test particle thickness $h$ is minimized with respect to the density of the test particle $\rho_p$. In other words, the optimum shape is that of a pancake where the test particle's mass is distributed over as large a surface area as possible. Additionally, the higher the air density, the stronger the force of the vertical wind, hence the smaller the difference between the speed of the test particle and the speed of the wind. Note that this discussion is only valid when considering directed flow, which is the continuum limit to zeroth order when neglecting lateral motions. These considerations are also complementary to discussion of a continuum drag coefficient.

The steady-state solution for equation (2.8) is $\mathrm{d}v/\mathrm{d}t = 0$. This corresponds to

$$-g + \lambda(w(z, t) - v(t))^2 = 0, \text{ where } \lambda = H(w - v)\frac{\rho(z, t)}{\rho_p h}, \tag{2.11}$$

giving, in the steady state,

$$v = w - \sqrt{\frac{g}{\lambda}}. \tag{2.12}$$

The negative square-root is selected as $v < w$ is required, and for $v > 0$ it follows that $\sqrt{\frac{g}{\lambda}} < w$. This gives the minimum wind needed to get the test particle to reach the steady state,

$$w \geq \left(\frac{g\rho_p h}{\rho(z)}\right)^{1/2}. \tag{2.13}$$

This condition is trivially the requirement that, for particles to move upwards, the upward velocity must exceed the terminal velocity. We will refer to this as the threshold velocity.

## (b) Estimates from vertical wind observations

We will examine three different test particles to illustrate the forces of vertical winds in the literature and make further hypotheses. We will call the first test particle our *standard test particle*, with general dimension of a nanometre (height and diameter) and standard dust density $1\,\mathrm{g}\ \mathrm{cm}^{-3}$ [62], giving it a mass of $\sim 3 \times 10^{-24}$ kg.

In figure 2, the condition equation (2.13) for our standard test particle and two biologically defined ones is presented. We used the NRLMSISE-00 model (see [64,65]) by implementing the Python module `fluids` and its class `atmosphere.ATMOSPHERE_NRLMSISE00`, to give the average air density, $\rho(z)$, as a function of altitude [66]. The change in the vertical velocity required for such a test particle to reach steady state is exponential and then linear. This corresponds to the change in the air density profile between the mesopause and thermosphere.

In our model, after some short transient period, the velocity of the test particle will become close to that of the wind, which is the threshold velocity. We now use the steady-state solution, equation (2.12), to make some simple estimates for the vertical distance a test particle could be carried upwards in the vertical winds reported by different observational studies.

Measurements of ion velocities can be considered as a proxy for neutral winds below altitudes of around 105 km. In [15], they used this technique to measure vertical winds in Greenland on two different nights at an altitude of about 103 km. As found in most other vertical wind data, the wind displayed oscillatory behaviour, as can be seen in figure 3, where it switched between upwards and downwards. The vertical winds range in magnitude between 10 and $50\,\mathrm{m\,s}^{-1}$, and on both nights there are long periods of consistent upward wind. If we consider a wind of $50\,\mathrm{m\,s}^{-1}$ for our small bacteria or bacteria organelle-sized test particle (density $2000\,\mathrm{kg\,m}^{-3}$, height 40 nm, radius approx. $2\,\mu\mathrm{m}$, mass $10^{-15}$ kg) and use the US Standard Atmosphere value for the air density at 100 km of $5.604 \times 10^{-7}\,\mathrm{kg\,m}^{-3}$ [59], we find that the upward velocity of the test particle is $13\,\mathrm{m\,s}^{-1}$

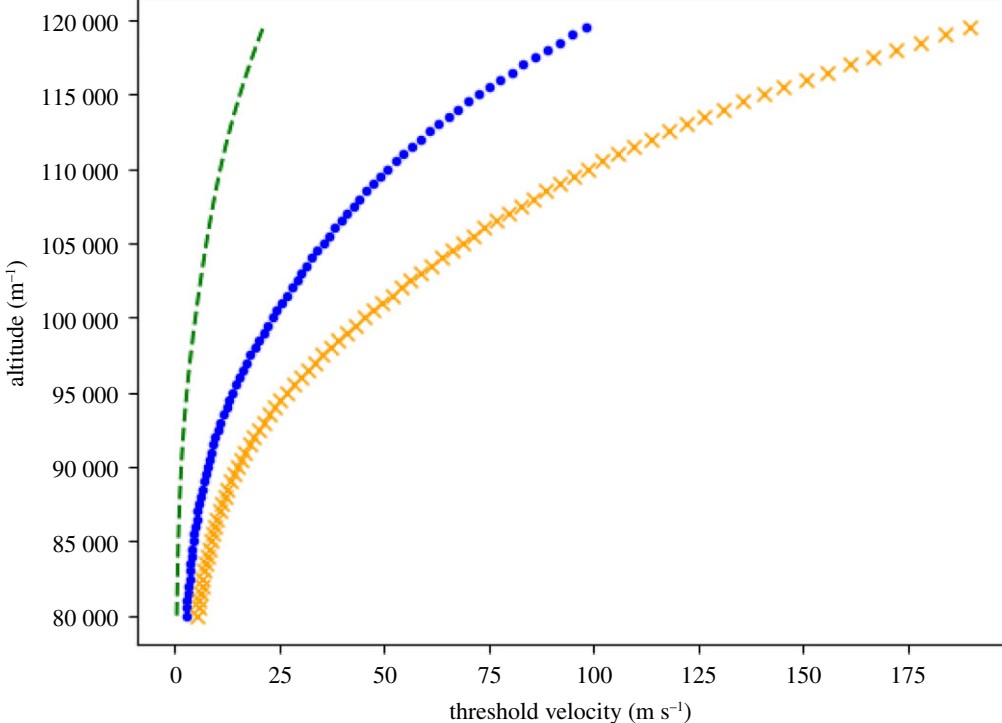

**Figure 2.** Threshold velocity equation (2.13) for three different test particles. Standard dust test particle of density 1000 kg m$^{-3}$, height and radius of a nanometre with a mass of $\sim 3 \times 10^{-24}$ kg (green dash). Virus-sized test particle of density 196 kg m$^{-3}$, thickness 109 nm (H1N1 virus from [63]) (blue dot). Small bacteria or bacteria organelle-sized test particle of density 2000 kg m$^{-3}$, height 40 nm, radius $\sim 2\,\mu$m and mass $10^{-15}$ kg (orange cross). (Online version in colour.)

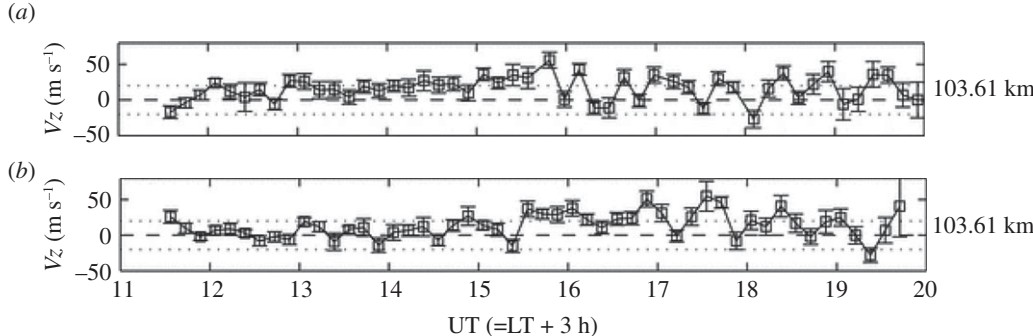

**Figure 3.** Time series of vertical ion velocities (neutral vertical wind proxy) at 103 km on 5 September (*a*) and 12 September (*b*) 2003. The dotted lines indicate 20 m s$^{-1}$. From [15]. (Online version in colour.)

(note that there will be a difference from the orange crossed curve in figure 2 as that uses a slightly different air density). Since the wind typically grows to a maximum over around 20–30 min let us suppose that the test particle on average has a vertical velocity of $\sim 7$ m s$^{-1}$. This would mean that the test particle would climb around 8.4 km in 20 min (figure 3).

The use of a disc geometry for the test particle means that the greatest forces on the test particle will occur when it maximizes its surface area and minimizes its thickness with respect to its mass. The stronger the vertical winds, the larger the test particle can be. The approximation is sufficient

to within an order of magnitude. However, the most significant approximation in our analysis is neglecting the horizontal winds. We justified this by the argument that the horizontal symmetry can be broken by the large fluctuations in vertical winds. These horizontal winds are between one and two orders of magnitude larger than the vertical winds in almost all situations.

The observed large vertical winds must have some extent in the horizontal. In studies of auroral arcs, vertical winds have been correlated over a distance of 300 km [17]. Fluctuation in the direction of the vertical winds, upwards or downwards, at these altitudes was also shown to correspond with the position of the auroral arc. They found in general that upward vertical winds were poleward of the auroral arc and downward winds equatorward of the auroral arc. In the horizontal, the neutral winds have been observed to change direction when auroral arcs appear such that they flow parallel in the region less than 50 km from the arc [67]. In [17], the effect of the horizontal winds on the vertical wind structure is discussed (paragraph 30) in a similar vein to the basic estimates we make here. They note that an air parcel rising at $30 \, \mathrm{m \, s^{-1}}$ will take 50 min to move from 140 to 240 km (these are the rough altitudes at which they correlated the vertical winds). However the horizontal winds over this altitude range have magnitudes in the range $200$–$500 \, \mathrm{m \, s^{-1}}$. They hypothesize that, when the horizontal wind is weaker, there may be higher correlation in the vertical wind in the horizontal and/or between these two different altitudes. Such discussions and analysis are effectively localized applications of the more general symmetry arguments we made at the beginning of this paper.

If we take the case of a lower magnitude horizontal wind, $200 \, \mathrm{m \, s^{-1}}$, then over the 20 min for a test particle to climb 8.4 km it will have moved 240 km in the horizontal, which is within the maximum correlation scale observed. If the horizontal wind magnitude is just $50 \, \mathrm{m \, s^{-1}}$ more then we reach this maximum observed 300 km correlation length. However, it could be argued that the correlation measurements are representative of the larger auroral arc, which would extend much further than 300 km. Observations of the horizontal winds and modelling do indicate that auroral arcs could form coherent neutral winds in the *E* emission airglow band in both the horizontal and vertical. More observational studies are required to determine these coherent properties.

There are many examples in the literature where we have sustained periods of vertical winds in the lower thermosphere which may or may not be related to auroral arcs (e.g. [19]). The mechanisms for many of these large vertical winds remain unknown [2]. Finally, let us take these estimates to the extreme by considering a test particle with dimensions such as those of the H1N1 virus, which has mass of around 0.8 fg and diameter of 109 nm, giving it a much lower density than bacteria of $\sim$196 kg m$^{-3}$ [63]. Note that for our disc model we assume that the diameter is equal to the height of the disc. The threshold velocity profile for such a test particle is plotted in figure 2 as the blue dotted curve. Then consider the case where this test particle is caught in a vertical wind of $50 \, \mathrm{m \, s^{-1}}$ along a long auroral arc. If we assume that the air density that the test particle experiences between 110 and 120 km is constant at $\sim 9 \times 10^{-8} \, \mathrm{kg \, m^{-3}}$ as it moves with the updraft, then the test particle will move upwards with a relative velocity of $\sim 2.5 \, \mathrm{m \, s^{-1}}$. At this speed for 1 h, the test particle will have moved up 9 km.

Vertical winds have been observed to remain upwards in a sustained manner for over an hour during geomagnetic storms, so if we imagine that the test particle were caught in the upward draft flowing along an auroral arc then it could be carried over 10 km upwards. The height it reaches simply depends on two properties: (i) the magnitude of the vertical wind relative to the air density and (ii) how long the Lagrangian trajectory of the test particle remains upwards with the necessary threshold velocity. By vertical winds alone, we argue that it is possible for a virus-sized test particle to climb to 120 km. However, we additionally suggest that altitudes greater than 120 km could be possible when multiple climbing factors (e.g. electrostatic levitation, photophoretic forces and vertical winds) act on a single test particle. We also note that there still remains a lot of uncertainty surrounding the large vertical winds in the 80–120 km altitude range, primarily because of a lack of consistent temporal and spatial observations, so perhaps even stronger winds are possible via extreme Joule heating. Once a particle has been lofted in an extreme vertical wind event it will experience significant horizontal transport, and will fall unless sustained by sufficient vertical wind or some other force such as those previously identified.

## (c) Considering downward winds: random walks

The problem with the current observations is that they give a limited slice of the Eulerian flow in the thermosphere. We believe our undeniably simple approach is valid assuming that these large updrafts form mesoscale Lagrangian coherent structures in which a test particle can be propelled upwards. These kinds of observational issues have been tackled with similar approaches in the troposphere for the transport of large dust particles over long distances [68]. In reality, the particle's path could be argued to follow some form of random walk in the auroral region, modulated by the solar activity, subsequent aurora and strength of the horizontal winds. The random walk approach may have considerable traction if more observations become available to establish patterns of these stronger vertical winds breaking the natural horizontal circulation symmetry.

In the horizontal, we have a reasonably continuous flow which is well documented and modelled. In the vertical the instantaneous large random winds appear as fluctuations, sometimes downwards and sometimes upwards or in our case sometimes not strong enough to support the weight of the particle. If we consider the Lagrangian motion of a particle, it will follow the Lagrangian coherent structure in the horizontal, which has been studied before [58]. The particle will move across above the surface of the Earth being pushed upwards and downwards by the local physics (e.g. Joule heating). For each moment of displacement in the horizontal direction, it will move either up or down in the vertical.

We considered framing the problem this way as it allows us to ignore the issue of the exact trajectory of the particle and leaves us with the classic statistical physics problem of the random walk in the vertical. This kind of simple application of stochastic physics is not new in atmospheric physics and has been successfully implemented for problems at the surface. However, a limiting factor is the air density at these high altitudes, which is too low to allow this mean field-theoretic approach as the threshold velocity condition is not met for the average or root-mean-square vertical velocities, which are between 10 and $20\,\mathrm{m\,s^{-1}}$ [69]. The approach we outlined may become useful as more field campaigns are conducted in the lower thermosphere, especially if auroral substorms can be better observed. Auroral substorms may exhibit vertical and horizontal wind correlations more conducive to this random walk method for particle transport.

Latitudes other than polar ones may well be important for transport via strong vertical winds but the evidence base for strong vertical winds outside of these regions is weaker, and the auroral arcs provide a ready-made Lagrangian coherent structure, which enables our 1D model to capture the basic picture of the dynamics reasonably. At lower latitudes, horizontal dynamics will be far more dominant owing to the lack of aurora. Only by further observation field campaigns of both high and low latitudes will it be possible to determine which is dominant for vertical transport.

There is also the question of how long such large particles would remain in the atmosphere. It has been argued that strong turbulence could keep particles in suspension for a longer time than is expected in the troposphere [70]. We suggest that, in our case, the mesopause would act as a buffer for particles above it because of the high turbulence in that region from breaking gravity waves and dynamic instabilities [71]. However, our interest is simply in how high an altitude the particles can reach. Since there exists a small but measurable concentration of micrometre-sized particles lower down in the mesosphere at any given time, it should imply that there will be some even smaller concentration at even higher altitudes.

## 3. Discussion

The purpose of this paper is to show using some simple estimates that particles larger than the air molecules can be lifted in the upper atmosphere, raising the possibility of biological particles being projected to higher altitudes than presently have been investigated. The estimates we present are just crude conjecture to an extent. However, it is all that can be estimated given the lack of observations and today's insufficient model capabilities without resorting to model

forcing. Nonetheless, this result has importance; we find that there is the possibility of these larger particles being carried from the upper mesosphere into the thermosphere. Our simple equation for the steady-state velocity of a particle blown upwards by the wind can be used by others to determine order of magnitude effects of vertical winds.

By showing that it is possible for large, heavy particles to reach these high altitudes, simply by vertical wind transport, interesting possibilities and questions are opened up. For example, it has been suggested that if biological particles can be found at a minimum altitude of 150 km, then hypervelocity space dust, which continuously impacts the atmosphere, has enough momentum to facilitate the planetary escape of such particles [72]. It was this work that also suggested vertical winds as a possible mechanism to facilitate the upward climb. Our estimates using reported observations of large vertical winds show that it is conceivable for such particles to be projected from near the highest measured altitudes in the mesosphere up to 120 km, which is 30 km off the minimum altitude given in the above reference.

The highest altitude that biological particles have been measured is 77 km [33]. Using specially adapted meteorological rockets Imshenetsky *et al.* [33] found bacterial and fungal organisms in the mesosphere between 48 and 77 km. To our knowledge, no further studies like this have been conducted since to push this biosphere boundary further. The recent analysis of swabs taken from the exterior of the ISS, which has an altitude of 400 km, does strongly indicate that particles of a biological origin (whole bacteria or DNA fragments) can reach deep into the thermosphere [34]. Our estimates support the hypothesis that these results are from the Earth's atmosphere. This is most likely in the form of DNA or organelle fragments owing to the size/mass constraints as the higher such particles can be projected into the thermosphere, the more they will be affected by the ascending branch of the global electric circuit, as Grebennikova *et al.* [34] suggested.

So we would suggest that biological particles can be found at altitudes higher than 77 km, especially considering the long history of large vertical winds measured in the upper mesosphere and thermosphere that occur during geomagnetic storms. The results of this paper contribute to the small but building body of evidence that the upper atmosphere should be of interest to the astrobiology community and that further experiment campaigns are needed. We suggest that searches for biological material could be conducted in the mesosphere and potentially the lower thermosphere.

Future work could consider the atmosphere of Mars, for which this vertical wind mechanism might be more suitable owing to the frequency of dust storms and the size of the dust. The weight of a particle on Mars is 38% less than that on Earth and the air density at the surface is only 1%. Dust storms have strong vertical winds and dust particles with radii larger than a micrometre [73,74]. Vertical particle transport in the Martian atmosphere is therefore likely to be strongly impacted by these storms, which can project dust up to around 80 km [73]. Smaller particles could attach to the dust and therefore reach higher altitudes. Given that micrometre-sized particles can be found at high altitudes around dust storms, then it is possible that biological particles of the same size could also be lofted to the same heights. If biological particles are present on Mars, and if such particles can be found at these higher altitudes, then they could be sampled and studied by satellites or probes such as balloons, without having to land on the planet's surface, using a methodology similar to that of [34,75].

Our results are complementary to the ideas associated with gravito-photophoresis, which has applications for climate science, as it has been suggested that particles could be engineered to reflect sunlight and be propelled by photophoretic forces [76]. Future work could examine combinations of thermospheric vertical winds and propulsion by photophoretic forces to determine the possible effects of geoengineering in the upper atmosphere. We think it is important to understand the possible impacts of any proposed geoengineering solutions on the upper thermosphere. This region has much more diverse chemical processes catalysed by the stronger solar radiation and ionospheric phenomena. So even if an engineered particle were to reach these high altitudes for a brief period, it could be long enough to create unexpected free radicals, which over gradual build-up could have unfortunate consequences for atmospheric composition, analogous to ozone depletion due to chlorofluorocarbons [77].

For the middle atmosphere community, the transport of particles into the upper mesosphere and lower thermosphere is usually related to the formation of the cold summer mesopause and noctilucent clouds, which are seasonal phenomena, as well as sudden stratospheric warmings (SSWs), which are intraseasonal events [78–80]. Our simple 1D model neglects cloud microphysics processes such as the charging of the particles, and their ability to be nucleation sites for mesospheric ice particles with subsequent formation of noctilucent clouds. One of the least understood aspects of the formation of the clouds is the nucleation rate [40]. One of the key components of this parameter is the vertical wind; therefore, the strong vertical winds will have an impact on the nucleation and sedimentation process. Stronger vertical winds will loft more particles to act as nuclei but they will also limit the growth time of the ice particles. Understanding the nucleation rate will greatly assist in setting bounds on transport mechanisms from the stratosphere into the thermosphere. These may have applications to long-term climate modelling [81].

## 4. Conclusion

The observations of large vertical winds reported for decades in the upper mesosphere and thermosphere have been used as the basis for developing a 1D vertical transport model for particles larger and heavier than the air itself. Our model contains two basic ingredients. First, it has the typical forces created by vertical winds. Second, it recognizes a symmetry that the profile of the vertical winds is basically similar over differing horizontal positions over the Earth. This symmetry thus allows the complicated horizontal dynamics to be factored out at the zeroth approximation, leaving only the 1D vertical motion. Exploiting this symmetry alongside the observed forces of these vertical winds makes this a very strong argument that our model does capture the essential features of the upward climb of particles by vertical winds. Any future more developed differential equation to that in this paper or any computer simulations trying to examine this problem in essence would need to capture the basic physics contained in our argument. These potential future developments could provide a more detailed portrayal of the vertical climb of test particles, but the basic features would be what we have argued here.

In the context of other observations and modelling studies of auroral arcs, our estimates indicate that such particles could indeed be transported deeper into the thermosphere than the previously measured and considered value of 77 km. We argue the case that a nanometre-sized particle could climb to an altitude of 120 km via vertical winds generated along auroral arcs. We call for further field campaigns to determine better the horizontal distribution of these large vertical winds and particularly between 90 and 150 km, as well as for modelling groups to consider examining 3D Lagrangian coherent structures in the thermosphere.

Data accessibility. Code with parameters to reproduce figure 2 can be seen and downloaded at: https://doi.org/10.5281/zenodo.3974863.

Authors' contributions. A.B.: conceptualization, investigation, methodology, supervision, writing—review and editing. D.J.B.: conceptualization, investigation, methodology, software, visualization, writing—original draft, writing—review and editing.

Competing interests. We declare we have no competing interests.

Funding. No funding has been received for this article.

Acknowledgements. A.B. thanks Jorgen Frederiksen for helpful discussions. D.J.B. acknowledges useful discussion about the representation of vertical winds in non-hydrostatic thermosphere models with Dr Jia Yue and Dr Katherine Garcia-Sage of the NASA Goddard Space Flight Center. A.B. is partially supported by STFC. D.J.B. is supported by the Carnegie Trust and STFC. We thank the two referees for their careful and insightful review of our manuscript.

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
