## [Peer Review File · Proceedings. Mathematical, Physical, and Engineering Sciences]

Review History

RSPA-2021-0626.R0 (Original submission)

Review form: Referee 1

Is the manuscript an original and important contribution to its field?

Excellent

Is the paper of sufficient general interest?

Excellent

Is the overall quality of the paper suitable?

Excellent

Can the paper be shortened without overall detriment to the main message?

Yes

Do you think some of the material would be more appropriate as an electronic appendix?

No

Do you have any ethical concerns with this paper?

No

Recommendation?

Accept with minor revision (please list in comments)

Comments to the Author(s)

The authors present an interesting and provocative hypothesis invoking observed vertical winds in the mesosphere and thermosphere to explain the possible existence of Earth biological particles in the thermosphere. As the authors explain, the reported observed large vertical winds are not yet represented in existing general circulation models. Taking in consideration the observed vertical velocities and assuming their existence for relatively large horizontal distance and in time scales of minutes to hours, the authors use a simple 1d model that explains altitude climbs of biological particles with masses larger than air. The model is far from being perfect, but it shows that if large vertical velocities like those reported are indeed real, Earth's biological particles could climb to thermospheric altitudes and perhaps leave Earth's atmosphere. Since demonstrating the validity of observed vertical velocities by other groups is not the aim of the paper, and the work motivates different research areas, e.g., understanding of observed vertical velocities, improvements of general circulation models to include such observations, astrobiological implications, geo-engineering, campaigns aim to measure biological particles in the mesosphere and thermosphere, etc., I recommend the paper to be published after considering the recommendations below.

- Title. Since the vertical velocities are invoke to justify the presence of biological particles at thermospheric altitudes, I suggest to modify the title to indicate that, something like: "On the force of vertical winds in the upper atmosphere to explain the existence of biological sized particles in the thermosphere" or similar ...
- Section 3 Motivation. Since "motivation" has been already included in the Introduction (Section 1), and some of the text is used to discuss the results, I suggest changing the section label to "Discussion" or similar.
- In page 13, L18-20 (geoengineering). Is your warming analogous to the Ozone hole and chlorofluorocarbons?
- General comments to consider in different parts of the text.
 - o Particles and biological particles are used in different parts of the text. I suggest to be explicit when it refers to biological particles. Recall that significant dust-sized particles are continuously being deposited in the upper atmosphere by meteoroids (e.g., Plane et al., 2015)
 - o Lidars and Incoherent scatter radars have observed signatures of metallic ion layers at altitudes as high as 180-200 km descending. Although their existence is associated to meteoroids, the high altitude concentrations are not expected to be from direct deposition, instead they might be due to some kind of transport (vertical winds?, parallel to B electric fields, etc.). Recent observations can be found in Chu et al (2021) and references there in. The authors might want to include such observations in their discussion.
 - o Variance of vertical velocities have been observed (and are expected) to increase with increasing altitudes. Recently Chau et al. (2021) have reported the existence of values as high as 60 m/s! These values are smaller than those at thermospheric heights, but they are extreme, more than 5 sigma larger than normal vertical velocities variability. Not only this constitutes additional evidence of high vertical velocities, but also it might extent the possibilities to non-aurora arc situations.
 - o I understand that the current speculation is limited to high-latitudes and during auroral-arc periods, which helps with the assumptions in the 1D model. If that is the case, could you please clarify? do you expect other latitudes and conditions to be included?
 - o Once the biological particles are at 150 km or above at high latitudes, do you expect significant horizontal transport? or suggested measurements of biological material should be conducted just at high latitudes?
- References
 - o Chau et al. (2021), <https://agupubs.onlinelibrary.wiley.com/doi/full/10.1029/2021GL094918>.
 - o Chu et al. (2021), <https://agupubs.onlinelibrary.wiley.com/doi/10.1029/2021GL093729>.
 - o Plane et al. (2015), <https://pubs.acs.org/doi/10.1021/cr500501m>.

Review form: Referee 2

Is the manuscript an original and important contribution to its field?

Good

Is the paper of sufficient general interest?

Good

Is the overall quality of the paper suitable?

Acceptable

Can the paper be shortened without overall detriment to the main message?

No

Do you think some of the material would be more appropriate as an electronic appendix?

No

Do you have any ethical concerns with this paper?

No

Recommendation?

Accept with minor revision (please list in comments)

Comments to the Author(s)

This article investigates the possibility of vertical transport of biologically-sized particles into the Earth's thermosphere (~90-600 km). This potential extension of the biosphere from the currently reported altitudes in the mesosphere (~ 80 km) to the thermosphere (~ 120 km) has implications for inter-planetary transport of biological samples through planetary escape of aerosols and for geoengineering through the transport and fate of aerosols in the middle- and upper-atmosphere (i.e., (i.e stratosphere-mesosphere and thermosphere respectively). The article presents evidence for significant vertical winds in the mesosphere and thermosphere, develops a vertical transport model for aerosols, and considers the structure of the wind systems that support the vertical transport model.

The article presents a scenario of how biologically active material can be transported high in the atmosphere, and highlights observations that may not be widely appreciated in the middle and upper-atmosphere communities. However, I have concerns about the evidence for vertical winds as currently presented, and the fate and transport of this material in the light of our understanding of the transport of other aerosols in the mesosphere.

The vertical transport of aerosols in the atmosphere will be of interest to the middle atmosphere community particularly in the study of the fate and transport of meteoric material, the meteoric metal layers, and noctilucent (polar mesospheric clouds, PMCs). The discussion of upward transport addresses both seasonal changes in the circulation (that result in transport of water vapor into the upper mesosphere and the diabatic circulation with the formation of the cold summer mesopause, and the formation of PMCs) and changes over periods of weeks (associated with events such as sudden stratospheric warming events (Randall et al., 2006; 2009; Meraner and Schmidt, 2016) . Investigations have focused on the ablation of meteors, formation of meteoric dust and the coagulation and growth of dust particles and their subsequent downward transport (Rapp and Thomas, 2006; Plane 2012; Wilms, 2016). Studies of mesospheric dust, detected when the dust becomes charged and detected as polar mesospheric summer echoes to radars, focus on their role as nucleation sites for mesospheric ice particles and the formation of PMCs. It is not clear in this environment how small aerosol particles will be transported through the upper mesosphere into the thermosphere without serving as nucleation sites for aerosol growth and sedimentation. Can the authors address if these processes could significantly impact

the transport of aerosols from the stratosphere into the thermosphere?

At times the discussion of the upward winds in the mesosphere and thermosphere implies that winds of 100 m/s can be found in the upper mesosphere. The wind measurements are primarily reported by interferometric measurements of the green (558 nm) and red (630 nm) lines in the thermosphere. The red line emissions come from the mid-thermosphere around 250 km, while the green line emissions come from altitudes near 130 km, the height of the emission can move in altitude based on the energy of the auroral precipitation (e.g., Billett et al., 2020). In the review by Smith (1998) the vertical winds on order of 100 m/s are associated with measurements in the mid-thermosphere from in-situ satellites and red-line observations. In their review, Larsen and Meriwether (2012) report upward winds of up to 50 m/s (180 km/h) in green line (Ishii et al., 2004), radar measurements at 103 km (Oyama et al., 2005), and in red-line measurements (Sipler et al., 2012). However, vertical winds of 100 m/s (360 km/hr) and larger are reported in the mid-thermosphere, rather than the lower thermosphere, based on measurements of the red line. Can the authors provide a clearer presentation of the observational evidence for the vertical winds and the altitude regions where they are found?

The discussion of the vertical winds by Kosch et al. (2010) in the article refers to their paragraph [30]. Paragraph 30 of Kosch et al. does not appear to discuss vertical winds. Can the authors clarify the results from Kosch et al., that they are citing?

As regards the transport of dust on Mars, the study by Vandaele et al. (2019) does not report dust at 80 km, but shows that during dust storms, that can last over weeks, there is vertical transport of dust to lower altitudes. Absorption of radiation by the lofted dust and subsequent heating results in changes in the general circulation that can transport water vapor up to 80 km. Can the authors clarify the results from Vandaele et al., they are citing. Furthermore, the authors note that the lower gravity on Mars will facilitate upward transport. However, the atmosphere on Mars is much thinner than the Earth's atmosphere. How will the combination of lower gravity (~ 38%) and lower air density (< 1%) on Mars yield vertical transport of aerosols?

References:

- Billett, D. D., McWilliams, K. A., & Conde, M. G. (2020). Colocated observations of the E and F region thermosphere during a substorm. *Journal of Geophysical Research: Space Physics*, 125, e2020JA028165. <https://doi.org/10.1029/2020JA028165>.
- Ishii, M., Kubota, M., Conde, M., Smith, R. W., and Krynicki, M. (2004), Vertical wind distribution in the polar thermosphere during Horizontal E Region Experiment (HEX) campaign, *J. Geophys. Res.*, 109, A12311, doi:10.1029/2004JA010657.
- Kosch, M. J., Anderson, C., Makarevich, R. A., Carter, B. A., Fiori, R. A. D., Conde, M., Dyson, P. L., and Davies, T. (2010), First E region observations of mesoscale neutral wind interaction with auroral arcs, *J. Geophys. Res.*, 115, A02303, doi:10.1029/2009JA014697.
- Lübken, F.-J., Berger, U., & Baumgarten, G. (2018). On the anthropogenic impact on long-term evolution of noctilucent clouds. *Geophysical Research Letters*, 45, 6681–6689. <https://doi.org/10.1029/2018GL077719>.
- Meraner, K., & Schmidt, H. (2016). Transport of nitrogen oxides through the winter mesopause in HAMMONIA. *Journal of Geophysical Research: Atmospheres*, 121, 2556–2570. <https://doi.org/10.1002/2015JD024136>
- Oyama, S., Watkins, B. J., Nozawa, S., Maeda, S., and Conde, M. (2005), Vertical ion motion observed with incoherent scatter radars in the polar lower ionosphere, *J. Geophys. Res.*, 110, A04302, doi:10.1029/2004JA010705.

Plane, J.M.C., (2012), Cosmic dust in the Earth's atmosphere, *Chemical Society Reviews*, 41, 6507-6518, doi: 10.1039/C2CS35132C.

Randall, C. E., Harvey, V. L., Singleton, C. S., Bernath, P. F., Boone, C. D., and Kozyra, J. U. (2006), Enhanced NO_x in 2006 linked to strong upper stratospheric Arctic vortex, *Geophys. Res. Lett.*, 33, L18811, doi:10.1029/2006GL027160.

Randall, C. E., Harvey, V. L., Siskind, D. E., France, J., Bernath, P. F., Boone, C. D., and Walker, K. A. (2009), NO_x descent in the Arctic middle atmosphere in early 2009, *Geophys. Res. Lett.*, 36, L18811, doi:10.1029/2009GL039706.

Rapp, M., & Thomas, G. E. (2006). Modeling the microphysics of mesospheric ice particles: Assessment of current capabilities and basic sensitivities. *Journal of Atmospheric and Solar-Terrestrial Physics*, 68, 715– 744.

Wilms, H., Rapp, M., & Kirsch, A. (2016). Nucleation of mesospheric cloud particles: Sensitivities and limits. *Journal of Geophysical Research: Space Physics*, 121, 2621– 2644. <https://doi.org/10.1002/2015JA021764>.

Decision letter (RSPA-2021-0626.R0)

23-Nov-2021

Dear Dr Brener,

On behalf of the Editor, I am pleased to inform you that your Manuscript RSPA-2021-0626 entitled "On the force of vertical winds in the upper atmosphere" has been accepted for publication subject to minor revisions in Proceedings A. Please find the referees' comments below.

The reviewer(s) have recommended publication, but also suggest some minor revisions to your manuscript. Therefore, I invite you to respond to the reviewer(s)' comments and revise your manuscript. Please note that we have a strict upper limit of 28 pages for each paper. Please endeavour to incorporate any revisions while keeping the paper within journal limits. Please note that page charges are made on all papers longer than 20 pages. If you cannot pay these charges you must reduce your paper to 20 pages before submitting your revision. Your paper has been ESTIMATED to be 17 pages. We cannot proceed with typesetting your paper without your agreement to meet page charges in full should the paper exceed 20 pages when typeset. If you have any questions, please do get in touch.

It is a condition of publication that you submit the revised version of your manuscript within 7 days. If you do not think you will be able to meet this date please let me know in advance of the due date.

To revise your manuscript, log into <https://mc.manuscriptcentral.com/prsa> and enter your Author Centre, where you will find your manuscript title listed under "Manuscripts with Decisions." Under "Actions," click on "Create a Revision." Your manuscript number has been appended to denote a revision.

You will be unable to make your revisions on the originally submitted version of the manuscript. Instead, revise your manuscript and upload a new version through your Author Centre.

When submitting your revised manuscript, you will be able to respond to the comments made by the referee(s) and upload a file "Response to Referees" in Step 1: "View and Respond to Decision Letter". Please provide a point-by-point response to the comments raised by the reviewers and the editor(s). A thorough response to these points will help us to assess your revision quickly. You can also upload a 'tracked changes' version either as part of the 'Response to reviews' or as a 'Main document'.

IMPORTANT: Your original files are available to you when you upload your revised manuscript. Please delete any redundant files before completing the submission process.

When uploading your revised files, please make sure that you include the following as we cannot proceed without these:

- 1) A text file of the manuscript (doc, txt, rtf or tex), including the references, tables (including captions) and figure captions. Please remove any tracked changes from the text before submission. PDF files are not an accepted format for the "Main Document".
- 2) A separate electronic file of each figure (tif, eps or print-quality pdf preferred). The format should be produced directly from original creation package, or original software format.
- 3) Electronic Supplementary Material (ESM): all supplementary materials accompanying an accepted article will be treated as in their final form. Note that the Royal Society will not edit or typeset supplementary material and it will be hosted as provided. Please ensure that the supplementary material includes the paper details where possible (authors, article title, journal name). Supplementary files will be published alongside the paper on the journal website and posted on the online figshare repository (<https://figshare.com>). The heading and legend provided for each supplementary file during the submission process will be used to create the figshare page, so please ensure these are accurate and informative so that your files can be found in searches. Files on figshare will be made available approximately one week before the accompanying article so that the supplementary material can be attributed a unique DOI. Alternatively you may upload a zip folder containing all source files for your manuscript as described above with a PDF as your "Main Document". This should be the full paper as it appears when compiled from the individual files supplied in the zip folder.

Article Funder

Please ensure you fill in the Article Funder question on page 2 to ensure the correct data is collected for FundRef (<http://www.crossref.org/fundref/>).

Media summary

Please ensure you include a short non-technical summary (up to 100 words) of the key findings/importance of your paper. This will be used for to promote your work and marketing purposes (e.g. press releases). The summary should be prepared using the following guidelines:

- *Write simple English: this is intended for the general public. Please explain any essential technical terms in a short and simple manner.
- *Describe (a) the study (b) its key findings and (c) its implications.
- *State why this work is newsworthy, be concise and do not overstate (true 'breakthroughs' are a rarity).
- *Ensure that you include valid contact details for the lead author (institutional address, email address, telephone number).

Cover images

We welcome submissions of images for possible use on the cover of Proceedings A. Images should be square in dimension and please ensure that you obtain all relevant copyright permissions before submitting the image to us. If you would like to submit an image for consideration please send your image to proceedingsa@royalsociety.org

Open Access

You are invited to opt for open access, our author pays publishing model. Payment of open access fees will enable your article to be made freely available via the Royal Society website as soon as it is ready for publication. For more information about open access please visit <https://royalsociety.org/journals/authors/open-access/>. The open access fee for this journal is £1700/\$2380/€2040 per article. VAT will be charged where applicable. Please note that if the corresponding author is at an institution that is part of a Read and Publishing deal you are required to select this option. See <https://royalsociety.org/journals/librarians/purchasing/read-and-publish/read-publish-agreements/> for further details.

Once again, thank you for submitting your manuscript to Proceedings A and I look forward to receiving your revision. If you have any questions at all, please do not hesitate to get in touch.

Best wishes
Raminder Shergill
proceedingsa@royalsociety.org
Proceedings A

Reviewer(s)' Comments to Author:

Referee: 1

Comments to the Author(s)

The authors present an interesting and provocative hypothesis invoking observed vertical winds in the mesosphere and thermosphere to explain the possible existence of Earth biological particles in the thermosphere. As the authors explain, the reported observed large vertical winds are not yet represented in existing general circulation models. Taking in consideration the observed vertical velocities and assuming their existence for relatively large horizontal distance and in time scales of minutes to hours, the authors use a simple 1d model that explains altitude climbs of biological particles with masses larger than air. The model is far from being perfect, but it shows that if large vertical velocities like those reported are indeed real, Earth's biological particles could climb to thermospheric altitudes and perhaps leave Earth's atmosphere. Since demonstrating the validity of observed vertical velocities by other groups is not the aim of the paper, and the work motivates different research areas, e.g., understanding of observed vertical velocities, improvements of general circulation models to include such observations, astrobiological implications, geo-engineering, campaigns aim to measure biological particles in the mesosphere and thermosphere, etc., I recommend the paper to be published after considering the recommendations below.

- Title. Since the vertical velocities are invoke to justify the presence of biological particles at thermospheric altitudes, I suggest to modify the title to indicate that, something like: "On the force of vertical winds in the upper atmosphere to explain the existence of biological sized particles in the thermosphere" or similar ...
- Section 3 Motivation. Since "motivation" has been already included in the Introduction (Section 1), and some of the text is used to discuss the results, I suggest changing the section label to "Discussion" or similar.
- In page 13, L18-20 (geoengineering). Is your warming analogous to the Ozone hole and chlorofluorocarbons?
- General comments to consider in different parts of the text.
 - o Particles and biological particles are used in different parts of the text. I suggest to be explicit when it refers to biological particles. Recall that significant dust-sized particles are continuously being deposited in the upper atmosphere by meteoroids (e.g., Plane et al., 2015)

o Lidars and Incoherent scatter radars have observed signatures of metallic ion layers at altitudes as high as 180-200 km descending. Although their existence is associated to meteoroids, the high altitude concentrations are not expected to be from direct deposition, instead they might be due to some kind of transport (vertical winds?, parallel to B electric fields, etc.). Recent observations can be found in Chu et al (2021) and references there in. The authors might want to include such observations in their discussion.

o Variance of vertical velocities have been observed (and are expected) to increase with increasing altitudes. Recently Chau et al. (2021) have reported the existence of values as high as 60 m/s! These values are smaller than those at thermospheric heights, but they are extreme, more than 5 sigma larger than normal vertical velocities variability. Not only this constitutes additional evidence of high vertical velocities, but also it might extent the possibilities to non-aurora arc situations.

o I understand that the current speculation is limited to high-latitudes and during auroral-arc periods, which helps with the assumptions in the 1D model. If that is the case, could you please clarify? do you expect other latitudes and conditions to be included?

o Once the biological particles are at 150 km or above at high latitudes, do you expect significant horizontal transport? or suggested measurements of biological material should be conducted just at high latitudes?

- References

o Chau et al. (2021), <https://agupubs.onlinelibrary.wiley.com/doi/full/10.1029/2021GL094918>.

o Chu et al. (2021), <https://agupubs.onlinelibrary.wiley.com/doi/10.1029/2021GL093729>.

o Plane et al. (2015), <https://pubs.acs.org/doi/10.1021/cr500501m>.

Referee: 2

Comments to the Author(s)

This article investigates the possibility of vertical transport of biologically-sized particles into the Earth's thermosphere (~90-600 km). This potential extension of the biosphere from the currently reported altitudes in the mesosphere (~ 80 km) to the thermosphere (~ 120 km) has implications for inter-planetary transport of biological samples through planetary escape of aerosols and for geoengineering through the transport and fate of aerosols in the middle- and upper-atmosphere (i.e., (i.e stratosphere-mesosphere and thermosphere respectively). The article presents evidence for significant vertical winds in the mesosphere and thermosphere, develops a vertical transport model for aerosols, and considers the structure of the wind systems that support the vertical transport model.

The article presents a scenario of how biologically active material can be transported high in the atmosphere, and highlights observations that may not be widely appreciated in the middle and upper-atmosphere communities. However, I have concerns about the evidence for vertical winds as currently presented, and the fate and transport of this material in the light of our understanding of the transport of other aerosols in the mesosphere.

The vertical transport of aerosols in the atmosphere will be of interest to the middle atmosphere community particularly in the study of the fate and transport of meteoric material, the meteoric metal layers, and noctilucent (polar mesospheric clouds, PMCs). The discussion of upward transport addresses both seasonal changes in the circulation (that result in transport of water vapor into the upper mesosphere and the diabatic circulation with the formation of the cold summer mesopause, and the formation of PMCs) and changes over periods of weeks (associated with events such as sudden stratospheric warming events (Randall et al., 2006; 2009; Meraner and Schmidt, 2016) . Investigations have focused on the ablation of meteors, formation of meteoric dust and the coagulation and growth of dust particles and their subsequent downward transport (Rapp and Thomas, 2006; Plane 2012; Wilms, 2016). Studies of mesospheric dust, detected when the dust becomes charged and detected as polar mesospheric summer echoes to radars, focus on their role as nucleation sites for mesospheric ice particles and the formation of PMCs. It is not clear in this environment how small aerosol particles will be transported through the upper mesosphere into the thermosphere without serving as nucleation sites for aerosol

growth and sedimentation. Can the authors address if these processes could significantly impact the transport of aerosols from the stratosphere into the thermosphere?

At times the discussion of the upward winds in the mesosphere and thermosphere implies that winds of 100 m/s can be found in the upper mesosphere. The wind measurements are primarily reported by interferometric measurements of the green (558 nm) and red (630 nm) lines in the thermosphere. The red line emissions come from the mid-thermosphere around 250 km, while the green line emissions come from altitudes near 130 km, the height of the emission can move in altitude based on the energy of the auroral precipitation (e.g., Billett et al., 2020). In the review by Smith (1998) the vertical winds on order of 100 m/s are associated with measurements in the mid-thermosphere from in-situ satellites and red-line observations. In their review, Larsen and Meriwether (2012) report upward winds of up to 50 m/s (180 km/h) in green line (Ishii et al., 2004), radar measurements at 103 km (Oyama et al., 2005), and in red-line measurements (Sipler et al., 2012). However, vertical winds of 100 m/s (360 km/hr) and larger are reported in the mid-thermosphere, rather than the lower thermosphere, based on measurements of the red line. Can the authors provide a clearer presentation of the observational evidence for the vertical winds and the altitude regions where they are found?

The discussion of the vertical winds by Kosch et al. (2010) in the article refers to their paragraph [30]. Paragraph 30 of Kosch et al. does not appear to discuss vertical winds. Can the authors clarify the results from Kosch et al., that they are citing?

As regards the transport of dust on Mars, the study by Vandaele et al. (2019) does not report dust at 80 km, but shows that during dust storms, that can last over weeks, there is vertical transport of dust to lower altitudes. Absorption of radiation by the lofted dust and subsequent heating results in changes in the general circulation that can transport water vapor up to 80 km. Can the authors clarify the results from Vandaele et al., they are citing. Furthermore, the authors note that the lower gravity on Mars will facilitate upward transport. However, the atmosphere on Mars is much thinner than the Earth's atmosphere. How will the combination of lower gravity (~ 38%) and lower air density (< 1%) on Mars yield vertical transport of aerosols?

References:

Billett, D. D., McWilliams, K. A., & Conde, M. G. (2020). Colocated observations of the E and F region thermosphere during a substorm. *Journal of Geophysical Research: Space Physics*, 125, e2020JA028165. <https://doi.org/10.1029/2020JA028165>.

Ishii, M., Kubota, M., Conde, M., Smith, R. W., and Krynicki, M. (2004), Vertical wind distribution in the polar thermosphere during Horizontal E Region Experiment (HEX) campaign, *J. Geophys. Res.*, 109, A12311, doi:10.1029/2004JA010657.

Kosch, M. J., Anderson, C., Makarevich, R. A., Carter, B. A., Fiori, R. A. D., Conde, M., Dyson, P. L., and Davies, T. (2010), First E region observations of mesoscale neutral wind interaction with auroral arcs, *J. Geophys. Res.*, 115, A02303, doi:10.1029/2009JA014697.

Lübken, F.-J., Berger, U., & Baumgarten, G. (2018). On the anthropogenic impact on long-term evolution of noctilucent clouds. *Geophysical Research Letters*, 45, 6681–6689. <https://doi.org/10.1029/2018GL077719>.

Meraner, K., & Schmidt, H. (2016). Transport of nitrogen oxides through the winter mesopause in HAMMONIA. *Journal of Geophysical Research: Atmospheres*, 121, 2556–2570. <https://doi.org/10.1002/2015JD024136>

Oyama, S., Watkins, B. J., Nozawa, S., Maeda, S., and Conde, M. (2005), Vertical ion motion observed with incoherent scatter radars in the polar lower ionosphere, *J. Geophys. Res.*, 110, A04302, doi:10.1029/2004JA010705.

Plane, J.M.C., (2012), Cosmic dust in the Earth's atmosphere, *Chemical Society Reviews*, 41, 6507-6518, doi: 10.1039/C2CS35132C.

Randall, C. E., Harvey, V. L., Singleton, C. S., Bernath, P. F., Boone, C. D., and Kozyra, J. U. (2006), Enhanced NO_x in 2006 linked to strong upper stratospheric Arctic vortex, *Geophys. Res. Lett.*, 33, L18811, doi:10.1029/2006GL027160.

Randall, C. E., Harvey, V. L., Siskind, D. E., France, J., Bernath, P. F., Boone, C. D., and Walker, K. A. (2009), NO_x descent in the Arctic middle atmosphere in early 2009, *Geophys. Res. Lett.*, 36, L18811, doi:10.1029/2009GL039706.

Rapp, M., & Thomas, G. E. (2006). Modeling the microphysics of mesospheric ice particles: Assessment of current capabilities and basic sensitivities. *Journal of Atmospheric and Solar-Terrestrial Physics*, 68, 715– 744.

Wilms, H., Rapp, M., & Kirsch, A. (2016). Nucleation of mesospheric cloud particles: Sensitivities and limits. *Journal of Geophysical Research: Space Physics*, 121, 2621– 2644. <https://doi.org/10.1002/2015JA021764>.

Board Member:

Comments to Author(s):

I am happy for this paper to be accepted if the points raised by the two referees are answered.

Decision letter (RSPA-2021-0626.R1)

01-Dec-2021

Dear Dr Brener

I am pleased to inform you that your manuscript entitled "On the force of vertical winds in the upper atmosphere - consequences for small biological particles" has been accepted in its final form for publication in *Proceedings A*.

Our Production Office will be in contact with you in due course. You can expect to receive a proof of your article soon. Please contact the office to let us know if you are likely to be away from e-mail in the near future. If you do not notify us and comments are not received within 5 days of sending the proof, we may publish the paper as it stands.

As a reminder, you have provided the following 'Data accessibility statement' (if applicable). Please remember to make any data sets live prior to publication, and update any links as needed when you receive a proof to check. It is good practice to also add data sets to your reference list.
Statement (if applicable):
Code with parameters to reproduce Figure 2 can be seen and downloaded at: <https://doi.org/10.5281/zenodo.3974863>

Under the terms of our licence to publish you may post the author generated postprint (ie. your accepted version not the final typeset version) of your manuscript at any time and this can be made freely available. Postprints can be deposited on a personal or institutional website, or a recognised server/repository. Please note however, that the reporting of postprints is subject to a media embargo, and that the status the manuscript should be made clear. Upon publication of the definitive version on the publisher's site, full details and a link should be added.

You can cite the article in advance of publication using its DOI. The DOI will take the form: 10.1098/rspa.XXXX.YYYY, where XXXX and YYYY are the last 8 digits of your manuscript

number (eg. if your manuscript number is RSPA-2017-1234 the DOI would be 10.1098/rspa.2017.1234).

For tips on promoting your accepted paper see our blog post:
<https://royalsociety.org/blog/2020/07/promoting-your-latest-paper-and-tracking-your-results/>

On behalf of the Editor of Proceedings A, we look forward to your continued contributions to the Journal.

Sincerely,
Raminder Shergill
proceedingsa@royalsociety.org